# A Feasibility Study of a Vibrotactile System Based on Electrostatic Actuators for Touch Bar Interfaces: Experimental Evaluations

**Taylor Mason [1], Jeong-Hoi Koo [1], Jae-Ik Kim [2], Young-Min Kim [3],* and Tae-Heon Yang [2],***

[1] Department of Mechanical and Manufacturing Engineering, Miami University, Oxford, OH 45242, USA; masontw@miamioh.edu (T.M.); koo@miamioh.edu (J.-H.K.)
[2] Department of Electronic Engineering, Korea National University of Transportation, 50, Daehak-ro, Chungju-si 27469, Korea; kji1023@ut.ac.kr
[3] Digital Health Research Division, Korea Institute of Oriental Medicine, 1672, Yuseong-daero, Yuseong-gu, Daejeon 34054, Korea
\* Correspondence: irobo77@kiom.re.kr (Y.-M.K.); thyang@ut.ac.kr (T.-H.Y.)

**Abstract:** Vibrotactile feedback is a very desirable feature for many touchscreen applications, creating a more engaging and effective user experience. Although it is common in small electronic devices, this feedback is often absent in large touchscreen devices because it is difficult to provide vibration sensations and control the magnitude throughout the display. Because of their long shape (over 20 cm), touch bar displays are susceptible to the same challenges that other large display types face. Thus, there is a need for a vibrotactile actuation system capable of generating a freely positionable and fully controllable point of stimulation with satisfying force output at any point of a touch bar display. This study proposes a new spring boundary condition vibrotactile system as a way to provide such feedback in touch bar interfaces. A prototype system was created using two electrostatic resonant actuators and a thin, narrow aluminum beam to study the effect of different actuator excitation parameters on the beam's response. By varying the number of actuators excited, magnitude, excitation frequency, and signal duration, a minimum vibration of 24.5 m/s$^2$ could be created in the beam, with the majority of the beam able to exceed 40 m/s$^2$. These results show that a targeted vibrotactile response at a given location in the beam can be achieved and sustained. This demonstrates a promising potential for generating a freely positionable and fully controllable point of vibrotactile stimulation at any point of a touch bar display.

**Keywords:** haptic feedback; vibrotactile feedback; haptic localization; freely positionable vibrotactile system; fully controllable vibrotactile stimulation

## 1. Introduction

Touchscreens have become increasingly popular in electronic devices due to their quick and intuitive use. As a means to provide a more engaging and accurate user experience, many touchscreen devices feature vibrotactile feedback. Vibrotactile feedback attempts to emulate physical interactions, such as mechanical buttons, textures, and friction, which are not typically present in touch displays [1]. This is especially beneficial for users that are visually impaired or have other disabilities [2,3]. Visual displays integrated with vibrotactile feedback have been shown in studies to improve user input speed, accuracy, and satisfaction compared to similar displays without such feedback [4–6]. Vibrotactile feedback can provide users with a variety of sensations depending on the intended application and on-screen visual feedback. In addition, vibrotactile feedback systems that can provide a variable feedback response strength can be used to convey different types of messages depending on the feedback magnitude. The benefits and applications of vibrotactile feedback are vast, making it highly desirable in many touch display applications.

Large touchscreen displays are currently being implemented in a multitude of applications, including automotive dashboards, information kiosks, tablet computers, and touch bars in laptop computers. Despite the many benefits offered by vibrotactile feedback, it is still missing in most large touchscreen displays. Many small mobile devices, such as smartphones and smartwatches, utilize a combination of visual, audible, and tactile feedback to convey information to their users. In order to provide tactile feedback in mobile devices, various vibrotactile actuators (such as eccentric rotary motors, linear resonant actuators, and piezo actuators) have been developed and commercialized [7–9]. However, this technology is not typically available in large touchscreen devices. This is because most haptic actuators are small and lightweight, being designed primarily for mobile devices. These actuators fail to create adequate vibrotactile feedback when applied to larger screens, often because of an insufficient vibration magnitude or inability to produce feedback throughout the screen.

Currently, the technology associated with electrostatics is being studied to provide haptic feedback for large touchscreen applications. Nakamura and Yamamoto proposed a visuo-haptic feedback system for providing haptic feedback to multi-users on a large touchscreen display using electrostatic haptic feedback and built-in capacitive sensors [10]. The sensation occurs when an electrically charged substance acts on one charged surface and a conductive fluid under the skin acts on the other [11]. Despite their simple structure and scalability for large display applications, electrostatic devices are immature for commercialization. Electrostatic display modules are limited in generating realistic haptic sensations, such as button click sensation, because they provide vibrotactile feedback when the finger slides on the touch surface [12]. Moreover, the performance of electrostatic touchscreens can be considerably affected by environmental factors, such as humidity and temperature [13,14]. Considering the limitations of existing haptic actuators and electrostatic display technologies, there exists a need to develop a new actuation system capable of generating a freely positionable and fully controllable point of vibrotactile stimulation with satisfying force output at any point in a large touchscreen display.

In addition to the need for developing haptic actuators geared toward large touch displays and interfaces, generating haptic sensations across large touch surfaces poses a challenge [15]. Controlling vibrotactile sensations at specific locations, or vibrotactile haptic localization, is being studied for small touchscreen applications [16,17]. This shows promising potential for use in small displays but is difficult to achieve in large displays. Although many large display vibrotactile systems attempt to effectively produce feedback throughout the display, there are often dead zones with no available feedback [18]. This is largely attributed to the edge boundary conditions used to secure touch displays. For example, a clamped boundary condition gives little control over the display's vibration mode shape without drastically changing the actuator's excitation frequency. Some of the most important factors that are desired for haptic localization are the ability to generate a response throughout the system, the ability to control the duration and magnitude level of the response, minimizing the number of actuators used, and having a quick response time.

Currently, controlling localized vibrotactile sensations in displays is done primarily using wave focusing, eigenfunction superposition, and traveling wave control methods. A study by Hudin et al. showed that time-reversal wave focusing can be used to control displacement impulses at one or several locations simultaneously [18]. This control method produces impacts in the supersonic range, well out of the range of vibrotactile sensitivity. However, it was shown that finger stimulation is still achieved through the ejection of the finger from the screen surface, which results in a larger finger displacement and duration compared to the screen. Lastly, this technique requires a large number of actuators for proper implementation and can only control pulse displacements.

An alternative method to wave focusing, called eigenfunction superposition, can also be used to create vibrotactile localization. This technique involves controlling modal coefficients and actuator weightings to render a targeted vibration distribution pattern. A study by Woo et al. describes using this principle to control the vibration in targeted "hot

zones" and suppress it in the other "cold zones" [19]. This method takes 18.2 ms to calculate the actuator weightings, showing a moderate response time. This study shows promise for the use of eigenfunction superposition for controlling vibration distribution patterns when dividing the screen up into large quadrants or grids but lacks the ability for high-resolution localization, which is the primary drawback of this control method. In addition, a large number of actuators were used to effectively utilize the eigenfunction superposition control method. Although eigenfunction superposition has important applications, the large number of actuators required and low-resolution localization make it not ideal for large touchscreen applications.

Another method used to create haptic localization is using traveling waves. Woo et al. also generated traveling waves as a means to create a targeted vibration distribution pattern [19]. They found that compared to eigenfunction superposition, using traveling waves was about 10% more successful (according to their metric) and had a 5.9 ms calculation time, roughly one-third as long. However, it was found that eigenfunction superposition uses approximately four times less power compared to the traveling wave control method. Using traveling wave control methods, it can be quite difficult to create long-duration vibrotactile feedback sensations. Similar to the superposition method, this method does not provide a high localization resolution despite also using a large number of actuators. A study by Ghenna et al. showed that eigenfunction superposition could be used to create an ultrasonic traveling wave to create variable friction in a vibrotactile display using only two transducers [20]. Traveling wave control methods have a multitude of uses, but they often require too many actuators to be practical in all touchscreen systems, and it is often difficult to create long and steady vibrotactile responses.

Most small modern touchscreen devices utilize vibrotactile feedback to communicate to users in a physical sense. However, this feedback is typically unavailable for devices with large screens. Long, narrow touchscreens are being considered for a multitude of applications, such as touch bars and automotive master switches typically located on the doors. As touchscreens continue to dominate the electronics market, there exists a need to develop vibrotactile feedback systems for all types of touchscreen devices. As an alternative to the previously discussed vibrotactile control methods, this study proposes a two-actuator spring boundary condition system for use in touch bar style displays. In contrast to a fixed boundary condition, using a spring boundary condition allows for more control and flexibility of the response in the display. The proposed control method allows for sustained controllable point of vibrotactile stimulation, as opposed to the impulse "button" sensations produced by other methods. Several different types of vibrotactile feedback sensations that offer more flexibility and control can be created using the proposed system.

In a study by Mason et al., a new haptic actuator based on electrostatic actuators, called the electrostatic resonant actuator (ERA), was developed for use in large touch displays [21]. Unlike typical electrostatic actuators, this actuator utilizes a moving mass and two electrodes to increase vibration intensity. The actuator produced a maximum peak to peak acceleration of 140.43 m/s$^2$ in the single-degree-of-freedom testing, showing promising potential for its use in generating vibrotactile feedback for large touch displays. Building on the previous work, this study aims to apply the ERA to narrow touch displays to provide a controllable vibration response of variable magnitude in large displays. In our study, we also demonstrated the possibility of providing broadband vibration patterns on large touch-sensitive displays by generating frequency pulses by inputting different frequencies to the two electrodes of the ERA [22].

The primary goal of this study was to experimentally evaluate the effects of using different actuator parameters in a narrow touch-bar-style display to study the vibrotactile localization potential and to evaluate the effectiveness of a direct single-degree-of-freedom actuator-to-touchscreen implementation. It is important to note that this study is solely limited to experimental evaluations to determine the feasibility of the vibrotactile actuation system. Future work is planned that includes extensive modeling and a simulation study. In this study, a two-actuator touch bar system was developed using modified dual-electrode

ERAs. A custom controller was designed and fabricated to control the actuation parameters of both actuators simultaneously. The system's vibration performance was experimentally evaluated to measure the effect of magnitude, excitation frequency, and excitation duration. To measure the vibration performance of the actuators on the system, peak acceleration measurements were taken along the length of the beam. The results were used to show that the vibration profile could be changed as desired to create a freely positionable, fully controllable vibration response. The next section presents the fabrication and the working principle of the two-actuator system. After explaining the fabrication of the system and the custom high voltage controller in Section 2, the paper presents the experimental setup and results. Then the experimental findings are discussed, showing that a fully controllable vibrotactile localization system can be achieved.

## 2. System Prototype and Controller Fabrication

To further study the dual-electrode ERA and work towards applying it to narrow touch displays, a preliminary touch-bar-style display was designed and fabricated. Figure 1a demonstrates the conceptual design of the haptic touch bar interface based on the dual-electrode ERA. The system is a proof-of-concept design, made primarily to study the effect of different ERA actuation parameters on the vibration response in the system. The system primarily consists of three main components: the left side dual-electrode ERA, the right side dual-electrode ERA, and the long, narrow touch display. As previously discussed, most large touch displays with vibrotactile feedback utilize a two-degree of freedom mounting method with clamped touchscreen ends, which can limit both the vibration amplitude and vibration profiles. To resolve this problem, this study proposes a mounting method that directly connects the actuator's oscillating mass to the display. To allow for this connection, the spring-mass actuator component was redesigned, as shown in Figure 1b. The modified actuator design features a central hole and welded M4 hex nut on the upper surface of the plate spring component. The radial beam springs of the central component have a stiffness of 16.18 kN/m. To achieve the desired total mass of 50 g per actuator, this design relies on the mass from a touch display as well as the nut and screw used to attach the actuator to the beam. Electrodes with holes in the center were used so as to not significantly interfere with the performance of the actuator and allow it to be directly connected to the beam. To amplify the performance of the actuator, a previously studied dual-electrode structure composed of an upper electrode and a lower electrode was applied.

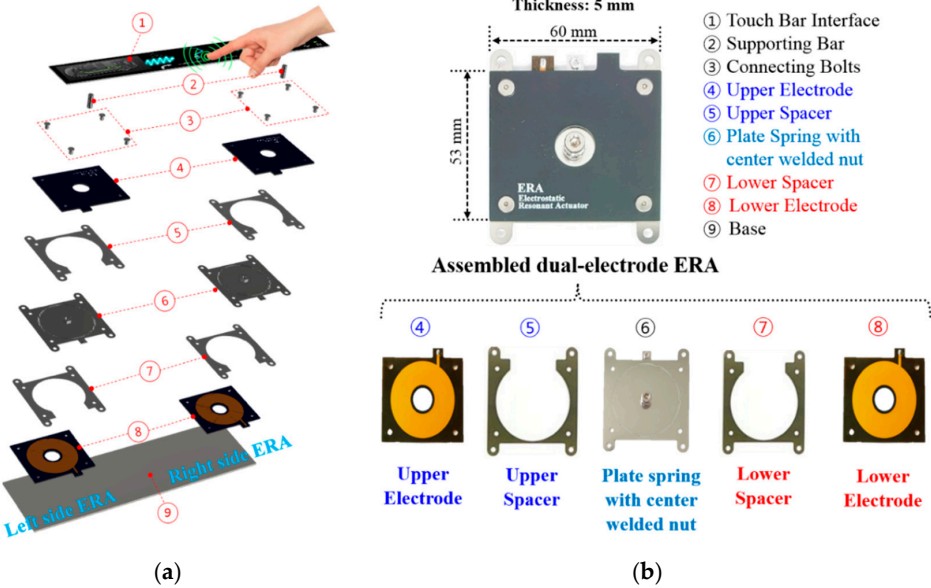

**Figure 1.** Haptic touch bar interface based on dual-electrode ERA: (**a**) An exploded perspective view of the touch bar interface design consisting of two dual-electrode ERAs and a long, narrow touch display; (**b**) assembled dual-electrode ERA and its components.

To control each ERA's actuation parameters simultaneously, a custom high voltage controller was fabricated. The controller can adjust one actuator's frequency, magnitude, phase delay, and signal duration independently from the other. This allows any combination of these parameters to be controlled simultaneously. The controller is also necessary to power the upper and lower electrodes within each actuator in an alternating fashion so as to never simultaneously create opposite attractive forces on the central vibrating component. The timing between each electrode activation must be accurate and consistent, making the controller highly important for proper function of the dual-electrode ERA. In addition to a controller, a high voltage amplifier must be used to amplify the control signal inputs. The amplifier takes the controller's low voltage input and significantly increases it, outputting the high voltage inputs required to properly power the electrodes within the actuator. Figure 2a shows the schematic structure of a circuit diagram for the controller and high voltage amplifier. This controller can power both actuators simultaneously with different parameters to produce separate signals. A total of 2 kV is required to power each electrode of the ERA module. A portable rechargeable battery is used as a power source to generate input signals of 2 V to the high voltage amplifier. Next, the high voltage amplifier increases the signal and outputs 2 kV to the electrodes. To produce high-voltage cosine signals with the desired frequency, high-voltage opto-diodes (OZ150SG, Voltage Multipliers Inc., Visalia, CA, USA) were adopted. The opto-diodes can adjust the high-voltage leakage according to the amount of infrared radiation. To produce a high-voltage cosine signal for one channel, two high-voltage opto-diodes are required for charging and discharging. Therefore, a total of four high-voltage opto-diodes are required for applying two high-voltage cosine signals to a single dual-electrode ERA module. Figure 2b shows the fabricated microprocessor-based controller. The output signal generated by the microprocessor passes through the sine wave generation elements and is converted into a waveform. After the signal passes through the current amplifiers, it is used as the input signal of the infrared LED that controls the opto-diodes. The high voltage DC output generated by the high-volt amplifier passes through the opto-diodes and is converted into a sine wave high voltage wave output to be applied to the ERAs.

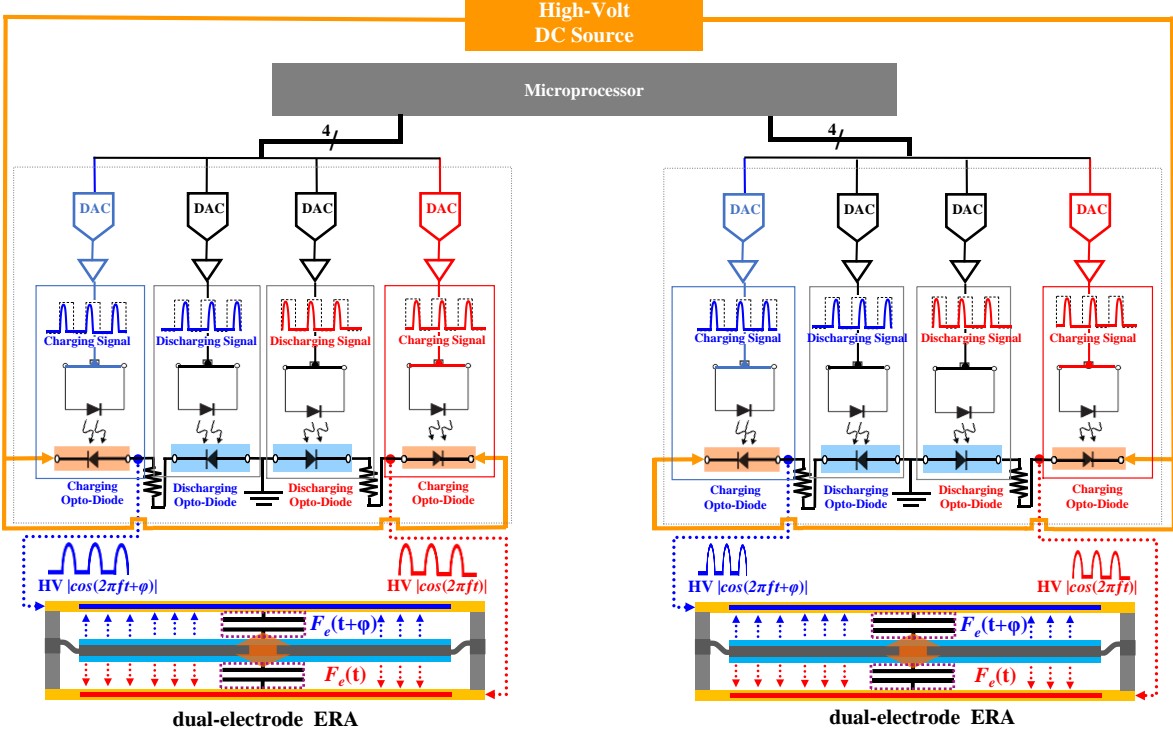

(**a**)

**Figure 2.** *Cont.*

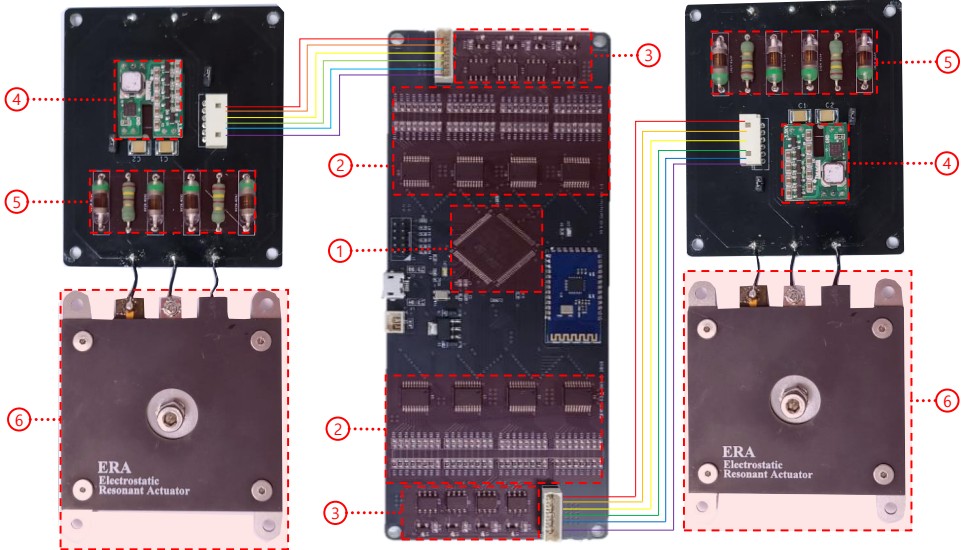

① Micro Processor
② Sine Wave Generation Elements
③ Current Amplifiers
④ High-Voltage Amplifier
⑤ High-Voltage Switching Elements (Opto-Diodes)
⑥ Electrostatic Resonant Actuator (ERA)

(**b**)

**Figure 2.** High-voltage controller for two ERAs: (**a**) Schematic of a circuit diagram capable of generating and controlling two channels of high-voltage signals; (**b**) fabricated microprocessor-based high-voltage controller and two connected ERAs.

## 3. Experimental Setup

The actuator-beam system developed for this experiment was made up of two actuators and a thin long, narrow aluminum beam, as seen in Figure 3a. The beam's dimensions are $305 \times 0.68 \times 29.9$ mm, with each actuator approximately 10 mm from a beam end. A hole was drilled in each side of the beam to allow the actuators to be bolted to it. Each actuator was secured to the beam identically using an M4 socket head cap screw, a washer, the M4 hex nut on the actuator, and an additional hex nut for spacing. This system was used to demonstrate how the vibrotactile signals produced by the actuators interact in a case comparable to one-dimensional. A vibrotactile signal is produced by each actuator, creating a vibration response in the beam. If both actuators are being excited, a mixed vibration response is created, depending on the actuation parameters of each actuator. Using this system, multiple experiments were conducted to better understand how the actuators' input parameters relate to the beam's output vibration response. To measure the vibration output transmitted to the aluminum beam, measurement points were marked on the beam, and an accelerometer was mounted on a measurement point to measure the vibration output. This acceleration measurement was performed using a validated accelerometer (Provider: PCB Piezotronics Inc., Model num: 352C66, Mass: 2 g, Type: ceramic shear ICP®accel., 100 mV/g, 0.5 to 10,000 Hz). The mass from the accelerometer likely affects the beam's response, meaning all acceleration measurements reported in this study represent the response with a small amount of additional mass at the measurement location. This is a potential limitation of this study, and the effect will be further evaluated in future testing. To control the actuation parameters of two ERAs simultaneously, a microprocessor-based controller and a tablet PC-based system equipped with haptic authoring GUI were prepared. As shown in Figure 3b, a haptic authoring GUI was developed to control the frequency, magnitude, phase delay, and duration of each of the two actuators.

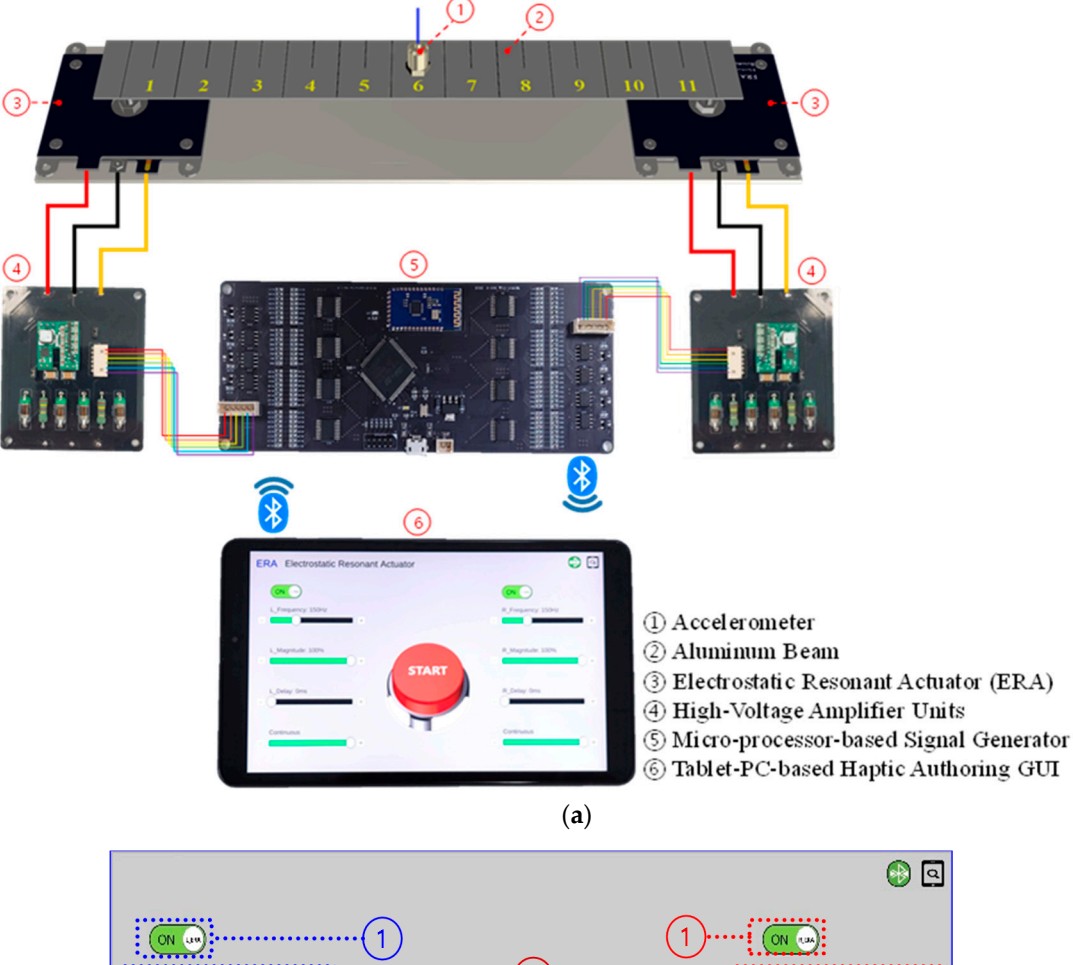

① Accelerometer
② Aluminum Beam
③ Electrostatic Resonant Actuator (ERA)
④ High-Voltage Amplifier Units
⑤ Micro-processor-based Signal Generator
⑥ Tablet-PC-based Haptic Authoring GUI

(**a**)

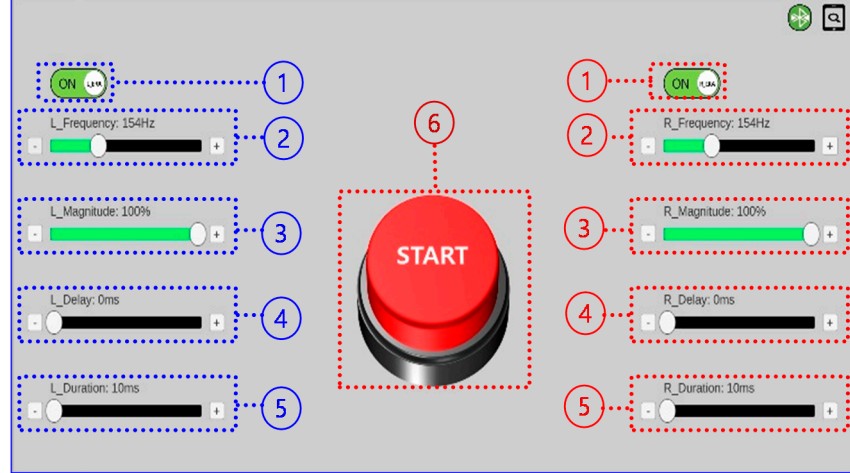

① Toggle Control: Left (CH1) & Right (CH2)
② Frequency Control: Left (CH1) & Right (CH2)
③ Magnitude Control: Left (CH1) & Right (CH2)
④ Phase Delay Control: Left (CH1) & Right (CH2)
⑤ Duration Control: Left (CH1) & Right (CH2)
⑥ Signal Transmission Button

(**b**)

**Figure 3.** Experimental setup showing: (**a**) Actuator-beam system and high-voltage controller; (**b**) haptic authoring GUI used to control actuator frequency, magnitude, phase delay, and duration.

## 4. Experimental Evaluations

This section presents the experimental evaluations of the system, comparing the vibration in the beam under different testing cases. This study proposes using different actuator vibration amplitudes, actuation frequencies, and signal durations as a means to control the vibration in the beam. The system's vibrotactile performance was experimentally measured in two studies: the single-actuator variable magnitude study and the dual-actuator pulse input study. These two experiments were collectively used to characterize the system and develop a means to control the vibrotactile feedback throughout the beam. In the future, this knowledge could be applied and extended to a large touchscreen plate with multiple actuators.

### 4.1. Overview

Using the experimental system and custom controller, the excitation parameters of each actuator could be simultaneously controlled to control the vibrotactile response in the beam. Through experimental testing, it was found that the natural frequency of the system differed depending on how many actuators were being used to excite the beam. The frequency range considered in this study was between 150 and 250 Hz, which is the range at which humans are most sensitive to vibrations [23,24]. When exciting the system using just one actuator, the natural frequencies within the desired haptic frequency range were found to be 171 and 196 Hz. When exciting the system using both actuators at the same frequency, the natural frequencies within the desired haptic frequency range were found to be 181 and 215 Hz. Both actuators were excited starting at the same time, so no phase difference between the actuators was present. Exciting the beam at any of these four natural frequencies with the proper number of actuators resulted in a specific vibration mode shape. However, it was found that exciting the beam at these frequencies using too many or too few actuators led to a cancellation effect. By exciting the actuators at one of the resonant frequencies and varying the number of actuators being excited, a switch from high to low vibration or low to high vibration could be achieved as desired. Figure 4 demonstrates the effect of using frequency and the number of actuators to control the vibration at an arbitrary point, in this case, 20 cm along the length of the beam.

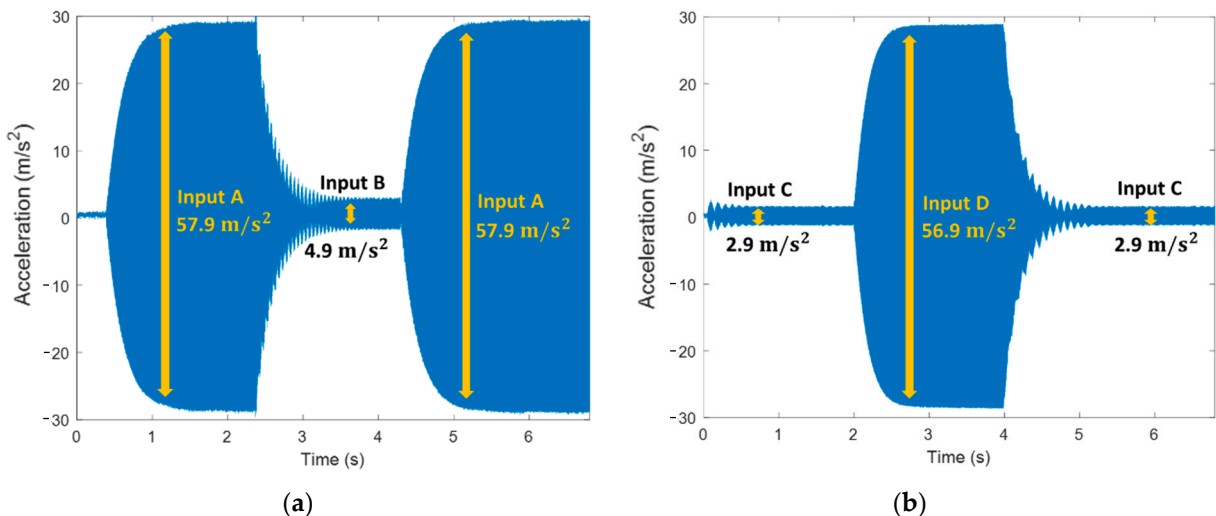

**Figure 4.** (**a**) Example showing an increase in vibration at one point in the beam; (**b**) example showing a decrease in vibration at one point in the beam.

In this example, multiple different actuator combinations with different input parameters were used, named Input A through Input D, for simplicity. In Figure 4a, the system is excited by Input A (the left side actuator is continuously excited at 196 Hz). After a short period, the system is excited by Input B for 2 s (both actuators are excited at 196 Hz).

Then the system returns to being excited by Input A. This results in a change in vibration from 57.86 to 4.90 m/s$^2$ and back to 57.86 m/s$^2$. In Figure 4b, the system is first excited by Input C (the left side actuator is continuously excited at 215 Hz). After a short period, the system is excited by Input D for 2 s (both actuators are excited at 215 Hz). Then the system returns to being excited by Input C. This results in a change in vibration from 2.94 to 56.88 m/s$^2$ and back to 2.94 m/s$^2$. This exemplifies how the vibrotactile feedback can be controlled in the beam using different actuator excitation parameters.

### 4.2. Effect of Actuator Magnitude

To be able to create a multi-level vibrotactile response, it was important to understand how the actuator magnitude affects the vibration in the beam. It was expected that lowering the magnitude of the actuators would proportionally lower the acceleration produced in the beam. To study the effect of magnitude, the acceleration was measured along the beam, keeping frequency constant and varying magnitude. Figure 5a shows the vibration along the beam, exciting it at 168 Hz with one actuator. The maximum peak to peak acceleration was 61.19 m/s$^2$, occurring at the center of the beam. Two node points are present around 10 and 20 cm. At these locations, the acceleration is below 9.81 m/s$^2$ regardless of the actuator's magnitude. As expected, decreasing the magnitude of the actuator lowers the magnitude of the response in the beam. Once the actuator reaches 50% magnitude at 168 Hz, it produces an insignificant response, with the acceleration below 9.81 m/s$^2$ throughout the beam. At this magnitude, the actuator's spring force is stronger than the electrostatic force, limiting output acceleration.

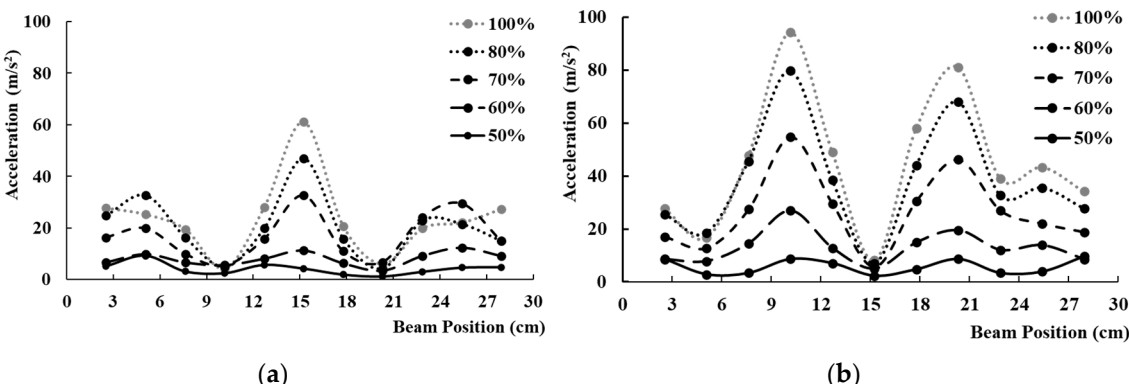

**Figure 5.** Acceleration along the beam, exciting a single actuator, varying the actuator's magnitude between 50 and 100% at a frequency of: (**a**) 168 Hz; (**b**) 196 Hz.

Figure 5b shows the vibration along the beam, exciting it at 196 Hz with one actuator. The maximum peak-to-peak accelerations that occurred were 97.77 and 86.59 m/s$^2$, occurring at 11.4 and 19.1 cm, respectively. Three node points are present around 6.4, 15.2, and 24.3 cm. At 15.2 cm along the beam, the acceleration is below 9.81 m/s$^2$ even when the actuator's magnitude was at 100%. At 6.4 and 24.3 cm, the acceleration is below 39.2 m/s$^2$, with the actuator's magnitude at 100%. As expected, decreasing the magnitude of the actuator proportionally lowers the magnitude of the response in the beam. Once the actuator reaches 50% magnitude at 196 Hz, it produces an insignificant response, with the acceleration below 9.81 m/s$^2$ throughout the beam.

### 4.3. Effect of Actuator Duration

It was shown that sustained vibrotactile feedback could be achieved using the proposed actuators and beam system but creating impulse "button" sensations was also desired. By running a continuous signal from one actuator then adding a short pulse signal from the other actuator, the vibration can be quickly increased or decreased, depending on the actuation frequencies and pulse duration. The maximum transient vibration response

was measured up to 200 ms after introducing the second signal. Figure 6a shows the vibration along the beam using 168 Hz signals at different pulse time durations. The maximum vibration occurs at 15.2 cm, with no pulse signal being added by the second actuator. By adding a pulse in the range of 10–200 ms, the vibration response can be decreased from 61.19 to 8.14 m/s$^2$, depending on the duration of the pulse.

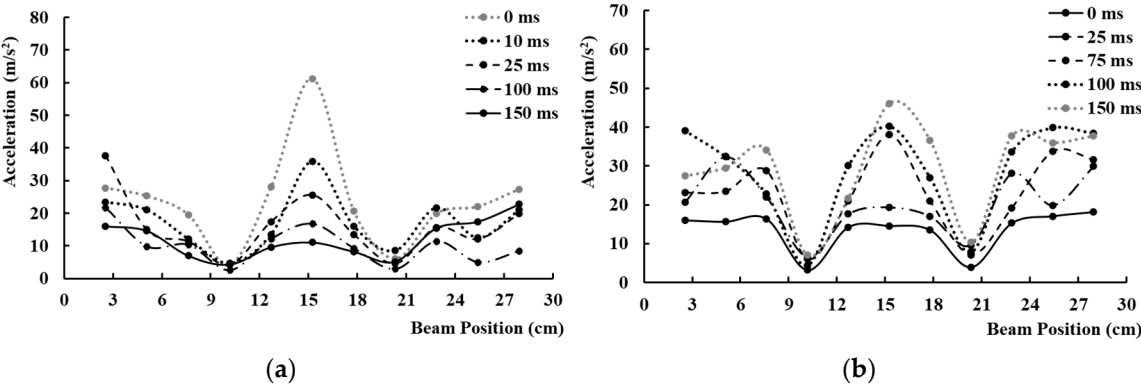

**Figure 6.** Acceleration along the beam, exciting both actuators at: (**a**) 171 Hz, varying one actuator's pulse excitation duration; (**b**) 181 Hz, varying one actuator's pulse excitation duration.

Using this same technique, the vibration signal can be increased when using 181 Hz actuation signals. At this frequency, adding a second actuator increases the vibration response in the beam, as seen in Figure 6b. When exciting only one actuator at 181 Hz, the maximum vibration is about 14.71 m/s$^2$. By adding a pulse in the range of 25–200 ms, the vibration response can be increased up to 44.13 m/s$^2$, depending on the location of the beam and duration of the pulse.

## 5. Discussion

Combining findings from the two previously discussed experiments, it can be shown that the vibrotactile feedback along the beam can be controlled, and a freely positionable vibrotactile feedback system can be achieved. Figure 7 shows the potential vibration profiles that can be achieved by changing excitation frequency. If a lower vibration magnitude is desired for a sustained period, then the excitation magnitude can be adjusted accordingly. If a higher or lower vibration magnitude is desired for a short period, then a pulse excitation signal can be used. Figure 7a shows the vibration response when exciting one actuator at its relevant natural frequencies. By changing the frequency, the desired mode shape can be achieved to account for the nodes and provide at least 29.42 m/s$^2$ (3 g) throughout most of the beam.

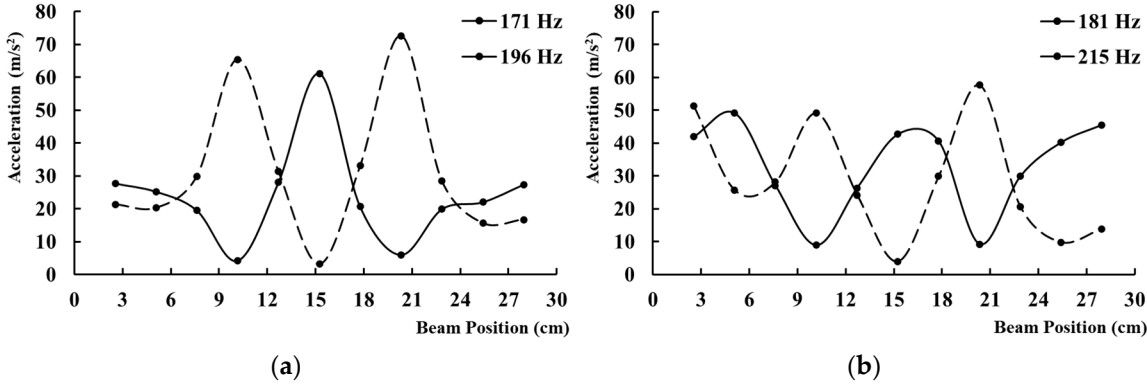

**Figure 7.** Unique vibration mode shapes that were experimentally measured when exciting: (**a**) a single actuator at 171 or 196 Hz; (**b**) both actuators at 181 and at 215 Hz.

Figure 7b shows the vibration response when exciting both actuators simultaneously, either at 181 or 215 Hz. By switching between these two similar frequencies, the location of the node can be accounted for and controlled. This results in a minimum vibration of 24.5 m/s$^2$ (2.5 g) in the beam, with the majority of the beam able to exceed 39.22 m/s$^2$ (4 g). Note that the system's output response frequencies are all close to the Pacinian corpuscles' most sensitive range and well above the minimum detection threshold [23,24].

Figure 8 demonstrates the potential of the touch bar actuation system. The solid black line represents the maximum achievable vibration for each point along the touch bar beam. Using each unique vibration mode shape, a freely positionable and fully controllable point of vibrotactile stimulation can be created throughout the touch bar display. To adjust the magnitude of a sustained response in the beam, the actuator's voltage magnitude can be adjusted as previously shown in the actuator magnitude experiment. To create a pulse increasing or decreasing response in the beam, the number of actuators excited can be adjusted as shown in the pulse experiment. By varying the number of actuators excited, the magnitude, excitation frequency, and signal duration, the desired vibrotactile stimulation can be generated at any point along the touch bar.

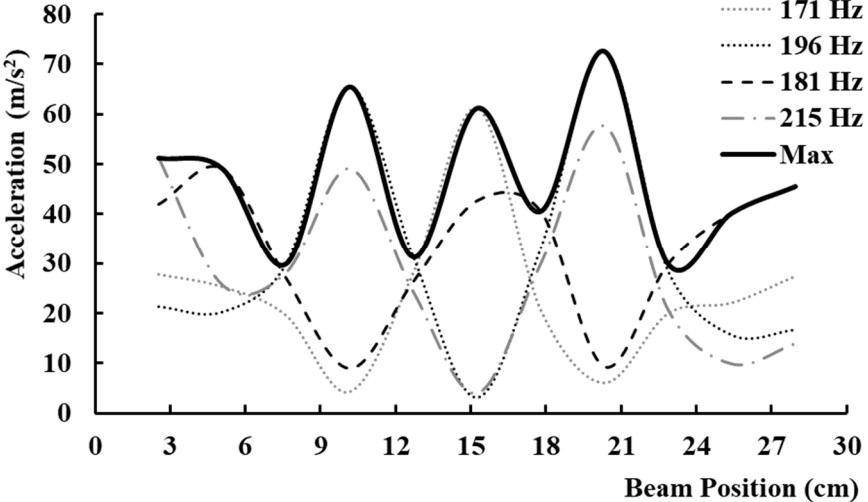

**Figure 8.** Experimentally measured maximum vibration potential of the freely positionable, fully controllable vibrotactile touch bar actuation system.

Although the direct actuator-to-display mounting method provides greater control over the display's mode shape, this can only be achieved by varying the frequency. In this system, vibration at a specific location is linked to actuation frequency. A change in frequency is required to generate vibrations at every location along the display. It is currently unclear if a user would be able to feel the different stimulation frequency because all four of the system's natural frequencies used in this study are close in frequency. Human trials are needed to identify the touch bar's feel and effectiveness.

## 6. Conclusions

In this study, a two-actuator system was proposed as a way to provide a freely positionable and fully controllable point of vibrotactile stimulation with satisfying force output throughout touch bar displays. A prototype system was assembled using two dual-electrode ERA and a thin aluminum beam. A custom controller and high voltage amplifier were fabricated to properly control both actuators simultaneously. Varying the number of actuators excited, actuator magnitude, excitation frequency, and signal duration, it was shown that the vibrotactile response in the beam can be controlled. The vibration node point locations can be moved without substantially changing the system's response by varying the actuator excitation frequency. Using a combination of actuator activations and excitation parameters, the vibrotactile response in the beam can be adjusted to achieve a

target response at a given location. These results show promising potential for the use of the ERA and direct actuator-screen mounting to provide controllable vibrotactile stimulations for touch bar displays. In future extensions of this study, the actuator will be applied to and tested upon full-size touch bar displays. In addition, this approach will be extended to full-scale large displays.

**Author Contributions:** Conceptualization, T.-H.Y., Y.-M.K. and J.-H.K.; methodology, T.-H.Y. and J.-H.K.; validation, T.M., T.-H.Y., J.-H.K. and Y.-M.K.; formal analysis, T.M. and J.-H.K.; controller and haptic authoring GUI development, J.-I.K.; resources, Y.-M.K. and T.-H.Y.; writing—original draft preparation, T.M.; writing—review and editing, T.-H.Y., Y.-M.K. and J.-H.K.; visualization, T.-H.Y. and Y.-M.K.; supervision, J.-H.K.; project admin-istration, Y.-M.K.; funding acquisition, T.-H.Y. and Y.-M.K. All authors have read and agreed to the published version of the manuscript.

**Funding:** This research was funded by a research grant from the Korea Institute of Oriental Medicine, grant number KSN2013110.

**Institutional Review Board Statement:** Not applicable.

**Informed Consent Statement:** Not applicable.

**Data Availability Statement:** Not applicable.

**Conflicts of Interest:** The authors declare no conflict of interest. The funders had no role in the design of the study; in the collection, analyses, or interpretation of data; in the writing of the manuscript, or in the decision to publish the results.

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
